# Cellular Plasticity and Tumor Microenvironment in Gliomas: The Struggle to Hit a Moving Target

**DOI:** 10.3390/cancers12061622

**Published:** 2020-06-18

**Authors:** Ricardo Gargini, Berta Segura-Collar, Pilar Sánchez-Gómez

**Affiliations:** Neurooncology Unit, Instituto de Salud Carlos III-UFIEC, 28220 Madrid, Spain; bertaseg@gmail.com

**Keywords:** glioblastoma, lower grade glioma, IDH mutations, EGFR, genetic and epigenetic alterations, glioma subtypes, tumor microenvironment, endothelium, pericytes, blood-brain-barrier, immune infiltrate, tumor-associated-macrophages

## Abstract

Brain tumors encompass a diverse group of neoplasias arising from different cell lineages. Tumors of glial origin have been the subject of intense research because of their rapid and fatal progression. From a clinical point of view, complete surgical resection of gliomas is highly difficult. Moreover, the remaining tumor cells are resistant to traditional therapies such as radio- or chemotherapy and tumors always recur. Here we have revised the new genetic and epigenetic classification of gliomas and the description of the different transcriptional subtypes. In order to understand the progression of the different gliomas we have focused on the interaction of the plastic tumor cells with their vasculature-rich microenvironment and with their distinct immune system. We believe that a comprehensive characterization of the glioma microenvironment will shed some light into why these tumors behave differently from other cancers. Furthermore, a novel classification of gliomas that could integrate the genetic background and the cellular ecosystems could have profound implications in the efficiency of current therapies as well as in the development of new treatments.

## 1. Introduction

The technological advances in the last two decades have produced a substantial improvement in our understanding of cancer biology. This has enabled the development of specific and effective therapies for several of the most common tumors, such as breast or prostate. However, these therapeutic designs have not been successful for most brain tumors, particularly for diffuse gliomas. This type of cancer remains one of the diseases with the worst prognosis of modern medicine. Although diffuse gliomas represent a small percentage of all primary tumors, they possess a mortality rate of 75% and belong to the top ten more deadly oncological malignancies, stressing the need to disentangle their molecular and cellular complexity [1,2].

Diffuse gliomas are neuroepithelial tumors. The two most common clinical forms are astrocytomas (the largest group) and oligodendrogliomas. In order to unify the diagnostic criteria, the World Health Organization (WHO) proposed a gradation system from grade II to IV based on the combination of parameters such as tumor mass extension, microvascular proliferation, genetic alterations, necrosis, and proliferation index. Graded II is assigned to slow developing tumors, whereas grade III tumors show histological symptoms of aggressiveness, such as cellular pleomorphism and mitotic activity. Both types of gliomas have an infiltrative nature and they are challenging for neurosurgeons. However, early surgical resection of these gliomas improves outcomes by delaying their transformation into higher grade tumors [3]. Grade IV gliomas, or glioblastomas (GBM), are the most aggressive tumors of the central nervous system and they normally show a fast pre and postoperative development. They appear primarily in adult individuals but also occur in children, and they represent the second leading cause of death from cancer. GBMs have astrocytic features, a high mitotic index and frequent areas of microvascular proliferation and necrosis [4]. The recent progress derived from different OMIC (genomics, transcriptomics, epigenomics, metabollomics) studies has added some molecular characteristics to the stratification of gliomas. The one that prevails over the histopathological classification is the identification of mutations in *isocytrate dehydrogenase 1/2* (*IDH1/2*), which defines gliomas with a better prognosis independently of their tumor grade [5]. Apart from containing different genetic alterations, the gliomas present different transcriptomic profiles, which discriminate between classic (CL), mesenchymal (MES), neural (N) and proneural (PN) tumors [6]. However, this stratification does not dictate a different therapeutic approach so it is not routinely performed in the clinic.

Glioma cells have a close relationship with their surrounding stroma, particularly with the vasculature. In order to progress into higher grade tumors, they produce several soluble factors that induce endothelial proliferation and shape the new vessels. Moreover, tumor cells contribute directly to the neo-vasculature through a vascular mimicry processes [7]. This complex interaction between tumor cells and the vasculature might explain why most antiangiogenic therapies based on single targets do not generate an increase in the survival of patients with aggressive gliomas [8]. Immune cells constitute another important component of the glioma microenvironment and they can reach up to 50% of the tumor mass content in some tumors. Recent pan-cancer studies have underlined the very special features of the immune compartment in gliomas, enriched in myeloid cells with strong immunosuppressive properties and a paucity of lymphocytes [9]. Furthermore, there is evidence of a close relationship between the vascular alterations and the immune component of gliomas. A wide variety of angiogenic molecules and cytokines exert a simultaneous effect on the endothelium and the tumor leukocytes. Moreover, changes in the function of the tumor vasculature affect the entrance of immune cells and/or their immunosuppressive properties; and the other way round, immune cells in the tumors can promote neovascularization [10,11]. However, the therapeutic consequences of this reciprocal regulation have not been seized yet.

Thanks to the effort of different consortia, adult gliomas are nowadays one of the best well-known tumors at the molecular level. However, these advances have not been translated into therapeutic benefits for the patients. We believe that the aggressiveness of gliomas, in particular GBMs, depends not only on the genetic alterations that they harbor, but also on the particularities of the brain tissue where they reside. Here, we will review some molecular and cellular aspects of gliomas, with a focus on the crosstalk between tumor cells and their surrounding stroma.

## 2. New Genomic and Epigenetic Insight of Diffuse Gliomas

### 2.1. Genetic Alterations that Define the Biology of Gliomas

After the 2016 revision of the WHO classification of brain tumors, the cIMPACT-NOW consortium has recently proposed a further integration of molecular data and histopathological features. Three main entities have emerged in adult diffuse gliomas, based on their mutational spectrum, histological characteristics, copy-number-variations and methylation patterns: (a) GBM, *IDH1/2* wild-type (IDHwt), grade IV; (b) astrocytoma, *IDH1/2* mutant (IDHmut), grades II–IV; and (c) oligodendroglioma, IDHmut, 1p/19q codeleted, grades II–III [12]. In addition, there is a percentage of astrocytomas (around 30%) and oligodendrogliomas (around 10%) without *IDH1/2* mutations. They have the epigenetic and genetic signature of GBMs IDHwt and have a worse prognosis than their mutant counterparts [5,13]. In some situations, molecular findings now prevail over the histopathology so that mixed or uncertain morphological categories can be better classified into specific entities. This is the case for the oligoastrocytomas, as detection of *ATP-dependent helicase* (*ATRX*) mutations (by loss of expression of the protein) indicates an anaplastic astrocytoma diagnosis, independently of the presence of oligodendroglioma features [14,15].

*IDH1/2* mutations are observed in 60–90% of lower grade gliomas (59–88% in diffuse astrocytomas and 68–82% in oligodendrogliomas) and in 10% of GBMs. The latter are now called grade IV-IDHmut astrocytomas and they are secondary tumors with frequent alterations in Tumor Protein P53 (*TP53)*, A*TRX* and *Capicua Transcriptional Repressor* (*CIC)* [5]. Patients with a grade IV IDHmut astrocytoma are younger and have a better prognosis, since the average survival after treatment is 31 months compared to 15 months in patients diagnosed with a GBM-IDHwt [16]. Whether the presence of *IDH1/2* mutations predicts a better response to chemotherapy remains controversial [17,18].

The IDHwt GBM group accounts for 90% of the grade IV cases and is mostly composed of primary tumors because they appear de novo, without signs on any previous lower grade lesion. They show a high frequency of genetic alterations in *Epidermal Growth Factor Receptor (EGFR),* and *Phosphatase and Tensin Homolog (PTEN)* [19]. In fact, detection of *EGFR* amplification or mutations (such as the common vIII deletion or point mutations) is associated with an increase in the aggressiveness of these gliomas, even if the histology corresponds to a grade III tumor [20]. Alterations that increase the signals activated by other receptors with tyrosine kinase activity (RTKs) or the loss of *PTEN* copies, which exacerbates the activity of these RTKs, have a similar effect [19,21].

Mutations in the *Telomerase Reverse Transcriptase* promoter (*TERT*p) have been found in 80% of gliomas, mostly in tumors with *EGFR* alterations. These mutations lead to upregulated expression of *TERT*, which codes for a catalytic subunit of the telomerase complex, thus increasing telomerase activity. This contributes to gliomagenesis by allowing cancer cells to overcome cellular senescence [22]. Notably, it has been shown that *TERT*p mutations are mutually exclusive with mutations in *ATRX*, as they have similar functions. Mutant ATRX can no longer inhibit the alternative lengthening of telomeres and therefore also promotes the immortalization of glioma cells [23].

Other common alterations in gliomas are *TP53* mutations and *Cyclin Dependent Kinase Inhibitor 2A and B (CDKN2A/B)* copy number losses, which show high frequencies in lower and higher grade gliomas, respectively [19]. The first ones are associated with the loss of control of the cell cycle and the gain of oncogenic function linked to the accumulation of mutant isoforms of p53 [24]. The latter are linked to the activation of the retinoblastoma (RB) pathway and to the proliferative niches that are observed in gliomas [25].

In general, the combination of different OMICs has established some predominant pathways that participate in the development of gliomas: RTKs, mitogen-activated protein kinase (MAPK), metabolic processes, interferon, and DNA-damage and cell cycle regulation (Figure 1) [19,26]. Alterations in these pathways tend to occur in an evolutionary way, with gain of chromosome 7 (*EGFR*, *platelet-derived growth factor A* (*PDGFA))*, loss of chromosome 9/9p (*CDKN2A/B*), and loss of chromosome 10/10q (*PTEN*) occurring at early stages in gliomagenesis [27,28]. By contrast, TERT promoter mutations occur later and confer a rapid growth advantage [28]. These main drivers cooperate with each other during glioma progression. Much of the information about these interactions has been gathered through the study of genetically engineered mouse models of gliomas, which enable us to explore them in the cell-of-origin [29,30]. All this knowledge has promoted the design of therapies to inhibit these pathways, generally by means of kinase inhibitors or monoclonal antibodies, although so far they have not generated the expected clinical benefits. A paradigmatic example is the failure of therapies aimed at inhibiting EGFR signaling [31]. The loss of double minute chromosomes carrying EGFR amplifications during treatment and their reappearance after therapy removal could partially explain the poor therapeutic potential of the kinase inhibitors [32]. Moreover, there is an evolutionary tendency to transform linear signaling pathways into networks with multiple regulatory options and redundant pathways [1,2,33]. In the particular case of EGFR, cooperation with inhibitors of downstream molecules or other RTKs has been proposed [34]. However, only some of the EGFR partners are known in advanced and resistance often comes from the activation of unexpected pathways [35]. Novel combinatorial treatments targeting multiple molecules within the same or parallel pathways are underway to overcome these limitations. In any case, further elucidation of the underlying biology of gliomas will be pivotal for developing more effective therapies.

### 2.2. IDH1/2 Mutations Have a Deep Impact in Glioma Pathology

As we mentioned previously, adult diffuse gliomas are categorized based on the presence of *IDH1/2* mutations, which is a strong prognosis indicator. The *IDH* genes are located on chromosome 2 and they are part of a family of genes (*IDH1*, *IDH2* and *IDH3*) that code for the isocitrate dehydrogenases. These homodimeric enzymes share structural characteristics but differ at the level of the intracellular localization: IDH1 is found in the cytoplasm while IDH2/3 are mitochondrial isoforms. All of them catalyze the oxidative decarboxylation of the isocitrate to α-ketoglutarate (α-KG), using as cofactors nicotinamide adenine dinucleotide phosphate (NADP+) (IDH1/2) or nicotinamide adenine phosphate (NAP+) (IDH3) and generating the reduced versions of these molecules: NADPH or NADH, respectively. These metabolites (NADH/NADPH) have been linked to suppression of apoptosis, stimulation of survival and cell growth [36]. Therefore, IDH enzymes, apart from being associated with energy synthesis (Krebs cycle), also contribute to cellular protection.

IDH mutations are somatic and mono-allelic and generate a change of an arginine residue located in the active center of the enzyme. The most frequent IDH1 mutated isoform found in gliomas (more than 85% of the cases) is the R132H, where an arginine is replaced by a histidine at position 132. A similar mutation is found in IDH2 (R172G/M/K) in 5% of IDHmut tumors. So far, no mutations related to cancer have been found in IDH3 [37]. Mutations in both IDH1 and IDH2 affect the critical residue for isocitrate binding and prevent the normal catalytic activity of the enzyme. Their pro-tumoral action is related in part to the reduction in NADPH levels and the decrease in levels of glutathione present in the body, which are necessary to reduce the deleterious effect of reactive oxygen species (ROS). In addition, IDH1/2 mutant proteins induce the production of an alternative metabolite, 2-hydroxyglutarate (2-HG) at the expense of α-KG. As a consequence, numerous enzymes dependent on the latter (histone demethylases, prolyl-hydroxylases and ten-eleven translocation (TET) enzymes) become inhibited, increasing the methylation of CpG islands and histones in the nucleus [36,37] (Figure 2). CpG islands contain a high concentration of cytosine and guanine pairs linked by phosphate. They constitute 40% of the promoters in human genes [38]. It has been shown that overexpression of mutant IDH1 in primary cultures of human astrocytes induces specific histone alterations that cause DNA hypermethylation and methylome remodeling, inducing the glioma CpG island methylator (G-CIMP) phenotype [39,40]. The G-CIMP phenotype is tightly linked with somatic IDH1/2 mutations and is normally associated with transcriptional gene silencing [41]. However, the transcription of certain genes that participate in the process of gliomagenesis, like *platelet derived growth factor receptor alpha* (*PDGFRA*) [42] and *microtubule-associated protein tau (TAU/MAPT*) [43], is actually induced after hypermethylation of their promoters. Altogether, these studies suggest that IDH1/2 mutations disrupt the chromosomal topology and orchestrate a complete change in the expression profile of the tumor cells, promoting tumor growth through the induction of oncogenic molecules [42] and the inhibition of pro-differentiation genes [44].

High hopes have been pinned on anti-IDHmut inhibitors and vaccines, which could have therapeutic utility in tumors with these molecular alterations [45,46]. However, these therapies could have long-term negative effects, given the relationship between *IDH1/2* mutations and the less aggressive behavior of gliomas. Although the reasons for this behavior are not known in detail, they have been associated with a decrease in cell proliferation [44]. Besides, 2-HG can inhibit hypoxia inducible factor 1, subunit alpha (HIF1α) signaling through the stimulation of prolyl-hydroxylases such as EGLN (egl-9 family hypoxia inducible factor 1) [47]. In relation with this, the epigenetic changes induced by IDH mutations has been linked to changes in the microenvironment that impair the progression of gliomas. These alterations will be discussed in further detail below.

### 2.3. DNA-Methylation Profiling in Diffuse Gliomas

The DNA-methylation profile of a tumor reflects its cell origin as well as the changes acquired somatically during tumor progression. The characterization of this methylome facilitates tumor classification in different cancers. It has proven to be a robust and reproducible technique, although it is still a slow and costly process. Moreover, there is limited knowledge on methylation regulatory mechanisms or how exactly the epigenetic changes affect the carcinogenic process. Despite all that, DNA-methylation changes are being used every day more as biomarkers at various stages of the cancer disease [48]. In the case of brain tumors, a recent comprehensive analysis has allowed the identification of almost 100 different entities across different age groups [49]. This study has shown the differences between the methylation patterns of IDHmut and IDHwt gliomas, confirming the similarities between IDHwt astrocytomas and GBMs. Moreover, it has underlined the difference between pediatric and adult gliomas. A previous analysis focused on gliomas revealed three DNA-methylation clusters among IDHwt tumors: RTK I, enriched in *PDGFRA* amplifications, RTK II, characterized by high frequency of chromosome 7 gain and chromosome 10 loss, as well as *EGFR* amplifications; and mesenchymal tumors, which show a methylation profile most similar to normal brain tissue despite substantial copy number changes. The three groups of gliomas have a similar clinical behavior regardless of the epigenetic differences [50]. A wider pan-glioma investigation allowed the subdivision of the IDHmut gliomas into a 1p/19q co-deleted (oligodendroglioma) cluster with elevated methylation (LGm3), a G-CIMP-high non-codeleted cluster (LGm2), and a newly-identified G-CIMP-low cluster with reduced methylation and a poorer overall survival when compared to the other two IDHmut tumors [26]. Other authors have confirmed the existence of three different subgroups among IDHmut gliomas, underlying the key role of other epigenetic modulators apart from mutant IDH1/2, like REST (RE1 Silencing Transcription Factor), in some of these subgroups [51]. These data demonstrate that DNA-methylation profiling (not only the G-CIMP phenotype) has a strong potential to classify gliomas and reflects the different trajectories followed by these tumors. Whether this can be used to predict the response to different treatments needs further evaluation.

### 2.4. The Definition of Glioma Subtypes: An Additional Step in the Knowledge of Pathology

Four subgroups of GBMs have been established based on their transcriptional profiles: PN, N, CL and MES [6,52]. This transcriptional classification can be applied to all diffuse gliomas: a big percentage of grade II and III gliomas fall into the PN category, although there are also tumors in the other three subclasses [53]. Each of these subgroups is characterized by specific genetic alterations, summarized in Figure 3. The N subtype is the most similar to normal brain tissue with increased expression of neuronal markers such as *NEFL*, *GABRA1*, *SYT1*, and *SLC12A5*. These tumors show a high degree of infiltration of normal cells and do not present a definite pattern of mutations, so this expression pattern is no longer considered associated with a specific subtype of gliomas [52].

The CL subtype is characterized by a high rate of cell division. In fact, some groups have named it as Proliferative [54]. These GBMs harbor chromosomal events that lead to the amplification of the *EGFR* gene (in 50% of cases with gene rearrangements), together with loss of *PTEN* and *CDKN2A* copies, whereas alterations in the *TP53*, *NF1*, *PDGFRA*, and *IDH1* genes are almost absent. CL GBMs show some response to radiation and conventional chemotherapies, probably due to the activation of the wild-type p53 function after the DNA damage induction. These tumors may also be sensitive to inhibitors of Mdm2, the negative regulator of p53. At the level of gene expression, the CL subtype shows elevated transcription of stem cells and progenitors-related genes, as well as Notch (*NOTCH3*, *JAG1*, and *LFNG*) and Sonic hedgehog (Shh) (*SMO*, *GAS1* and *GLI2*) signaling molecules [6,52,54].

The expression profile of MES gliomas is associated with that of mesenchymal cells, as well as with angiogenesis processes. They overexpress *CHI3L1/YKL40*, *MET, CD44* and *MERTK* genes, astrocytic markers and members of the *TNF* superfamily, including the *NFkB* pathway [55,56,57]. Inactivation of *NF1* (37%), *TP53* (32%), and *PTEN* (32%) genes are frequent in these gliomas. This GBM subtype shows a relative response to intensive treatment with temozolomide (TMZ) but a high resistance to radiotherapy, linked to the ability to NFκB (nuclear Factor kappa-light-chain-enhancer of activated B cells) activation and changes in their tumor microenvironment. Moreover, they could be sensitive to Ras and phosphoinositide 3-kinase (PI3K) inhibitors, as well as to certain anti-angiogenic molecules [6,52,54,55,56].

The PN subtype shows a better prognosis compared to the other groups. These tumors show an expression profile reminiscent of neuronal development, with high expression of known oligodendrocytic (*PDGFRA*, *OLIG2*, *TCF3*, and *NKX2-2*) and neural (*SOX2/9, DCX*, *DLL3*, *ASCL1*, and *TCF4*) markers. PN tumors appear in younger patients and they carry frequent mutations in the *IDH1* gene (30%). They often overexpress PDGFRA, either by epigenetic activation of its promoter or by amplification/mutation of the *PDGFRA* gene (35%). They also present frequent mutations in *TP53* (54%) and *PIK3CA/PIK3R1* (19%) and significant accumulation of chromosome 7 amplification and loss in chromosome 10 (>50%), although the latter event is more frequent in the CL subtype. PN gliomas are highly invasive and show less sensitivity to aggressive cycles of radiotherapy and TMZ [6,52,54], which in the first case has been directly linked to the epigenetic upregulation of the DNA repair response in the presence of IDH mutations [58]. The PN subgroup is the only one that includes secondary high-grade gliomas. Notably, PN tumors without *IDH* mutations have a survival comparable to that of other expression subtypes [53].

A recent analysis of the molecular subtypes of the diffuse gliomas using tumor maps (generated combining the mRNA and the methylation profiles) has allowed the identification of specific clusters in the different subtypes (Figure 3). The graph indicates that there are two divergent patterns in the PN and the CL subgroups. The neural tumors appear more dispersed, probably because this subclass does not reflect a specific history of the cells, rather a contamination of the tumor sample with normal tissue. It is noteworthy that MES GBMs do not cluster together and, in many cases, it is difficult to separate them from the CL tumors, suggesting a blurred frontier between these two entities.

The molecular classification of gliomas reflects their biology, suggesting that tumors with similar expression profiles would present alterations in genes that participate in the same signaling pathways. Molecular clustering should then facilitate the accurate prediction of the response of gliomas to inhibitors of these specific pathways and thus the allocation of patients into different clinical trials. However, caution should be taken as gliomas in general, and GBMs in particular, represent a highly heterogeneous disease. There are evidences showing an inherent variability in the expression of the diverse transcriptional programs inside any given tumor [59]. Moreover, different clones in one single tumor might have different drug sensitivity [60], making exceedingly difficult to predict the response of a particular GBM based on its transcriptional profile. Intratumor DNA methylation heterogeneity has also been described [61], although the methylation subtypes seems to be more consistent during tumor progression [62]. Future studies with more refined and robust profiling are needed. We propose that a combination of transcriptional, epigenetic and genomic data, especially at the single-cell level, will offer a better classification of gliomas that could have a predictive value.

## 3. Heterogeneity of Glioma Cancer Stem Cells

The cancer stem cells have been defined based on their similarities with progenitor cells in the different tissues, along with their capacity to repopulate the tumors after treatment. They have been isolated from different types of primary tumors, including high grade gliomas [63]. The glioma stem cells (GSCs) are highly resistant to the DNA damage inflicted by traditional therapies like radiotherapy and TMZ. They are usually in a quiescent state, which makes them less susceptible to these agents. Furthermore, it has been observed that they express membrane transporters of the ABC type (ATP-binding cassettes) that expel chemotherapeutic agents, and the cytosolic enzyme ALDH (aldehyde dehydrogenase), which acts as a detoxifier [64,65]. In addition, the pathways associated with the maintenance of the stem-like phenotype such as HIF1α, Notch, Wnt (Wingless-type MMTV integration site) or Shh are activated in GSCs. These molecules increase different survival signals and make them more resistant to cytotoxic therapies [66]. As a proof-of-concept of the relevance of stem cells in gliomas, chimeric antigen receptor T cells (CAR-T) recognizing CD133, a protein coded by the prominin I gene that is highly expressed in GSCs [63], has demonstrated superior efficacy in patient-derived GBM xenograft models [67].

The details of the differentiation and maintenance of GSCs have not been fully described but both autocrine and paracrine cues derived from the surrounding stroma are involved. An interplay between tumor cells and their vascular niche has been described: GSCs modify the tumor vessels and promote the formation of new ones. In return, the vascular niche provides the metabolic conditions and the adequate extrinsic signaling for the maintenance of the tumor stem cells [68,69,70]. More recently, a similar bi-directional interaction has been described between GSCs and other tumor ecosystems, enriched in hypoxic or immune signals [70,71]. All these results challenge the idea of a GSC identity. Alternatively, it has been proposed that the stemness features of gliomas are not linked to a particular cell type but rather to a particular cellular state that most cells in the tumor can adopt depending on their genetic background and the instructions from the microenvironment. Thus, the tumorigenic potential of GBM cells would be linked to an intrinsic plasticity of the tumor cells and not to the multipotentiality of a particular small group of GSCs [72]. The intrinsic and extrinsic regulators of this cellular plasticity contribute to the heterogeneity of gliomas.

In the same way that different GBM subtypes have been identified through genetic studies and gene expression analysis, several molecular studies have described the heterogeneity of GSCs. Most of these studies categorize these cells into PN or MES GSCs, each one carrying different molecular alterations responsible for their phenotype. Notably, no GSCs of the CL subtype has been convincingly identified ex vivo. PN GSCs express genes involved in neurological development while the expression of genes related to the extracellular matrix, such as integrins, is enriched in MES GSCs. There are common markers for both types of GSCs such as *NESTIN* but many others are differentially expressed. PN GSCs show high transcript levels of *CD133* and the Notch signaling pathways. By contrast, high levels of expression of CD44 and the absence of CD133 characterize MES GSCs [73,74,75,76]. PN GSCs have a greater ability to invade the brain parenchyma when injected into brains of immunocompetent mice, which has been associated with the increased expression of *OLIG2* [77]. By contrast, MES GSCs generate more aggressive tumors and have a greater angiogenic capacity [68,76]. Likewise, the MES GSCs have a greater cellular plasticity, modulated by their interaction with the microenvironment and governed by the epithelial-to-mesenchymal transition (EMT), also called glial-to-mesenchymal transition in gliomas. EMT is a dynamic and reversible process in which cells lose their ability to polarize and form intercellular junctions. Instead, they acquire mesenchymal characteristics, such as elevated resistance to apoptosis and high migratory capacity. This increased plasticity is linked to a greater tumorigenic capacity [75]. Furthermore, it has been recently proposed that EMT allows tumor cells to differentiate into pericytes (mural cells of a mesenchymal lineage), which promote angiogenesis and neovascularization of gliomas [43,78,79,80]. One of the main regulators of this EMT process is the transcription factor TAZ (Tafazzin) or WWTR1 (WW domain-containing transcription regulator 1), which regulates cell proliferation, migration and invasion in cancer cells [43,75,81]. In fact, recent studies have demonstrated the exclusive expression of TAZ in the MES GSCs and the absence of expression in the PN GSCs [82], linking this transcriptional regulator with the maintenance of mesenchymal stem cell properties.

Some recent studies combining single-cell sequencing approaches and bulk genetic and in silico analysis have further contributed to describe the phenotypes of GSCs that reside on IDH wild-type GBMs. Wang and co-workers [83] have shown that proliferating GBM cells can be described by a single gene signature axis with expression of canonical PN and MES markers in both extremes. These two types of cells, their progeny (differentiated astrocytes and oligodendorocytes, and the stroma (vascular and immune cells) are sufficient to explain the heterogeneity of GBMs [83]. Another study has identified four tumor cell states (not GSCs): astrocytic-like, which represent a big percentage of the cancerous cells in CL GBMs, mesenchymal-like, abundant in MES GBMs, and neural and oligodendrocyte progenitor-like, present at different rates in PN GBMs. The genetic status of *CDK4*, *EGFR* and *PDGFRA* loci, under the influence of the tumor niche, drives the different states in a plastic way [84]. Although the clusters of expression vary depending on the authors, all the studies agree in their plastic nature and confirm that the human brain microenvironment is essential to recapitulate the heterogeneity observed in the tumors [85]. There are evidences showing that phenotypic plasticity is promoted during tumor progression and in response to therapy, with a tendency to change from PN to MES subtypes [55,86] and a frequent loss of the CL signature [62]. Thus, a holistic view of gliomagenesis can be proposed, one that integrates the genetic and epigenetic drivers, the cell of origin and the microenvironment of the different GSCs (or the different cellular states), which support their growth and/or survival. All these components determine the molecular phenotype of the tumors and orchestrate their evolution in a fluid manner, allowing them to escape from the different treatments. In that sense, it has been recently shown that combinatorial therapies that target the pathways enriched in the different GSCs, for example inhibiting EGFR, FGFR3 (fibroblast growth factor receptor 3), or Wnt (MES targets) and Survivin (PN) [83], or by using BMI (B Lymphoma Mo-MLV Insertion Region 1) (MES) and EZH2 (Enhancer Of Zeste 2 Polycomb Repressive Complex 2) (PN) inhibitors [87] are more effective than approaches directed towards only one of the GSC subtypes. Therefore, plasticity of the tumor phenotypes is an intrinsic property of aggressive gliomas that presents unique challenges (and opportunities) for treatment.

## 4. Glioma Microenvironment: Distinct Vascular and Immune Compartment

Pan-cancer analyses have revealed that GBMs are one of the most aggressive forms of cancer [88]. These studies have measured the mutation rates and the frequency of aneuploidies and alterations in oncogenic pathways and they have shown that GBMs do not accumulate more genetic alterations than other tumors [89,90,91]. Consequently, the extremely bad behavior of GBMs cannot be sustained in molecular alterations alone. Alternative, it has been proposed that the special characteristics of the brain tissue fuel the development of an aggressive tumor. It is noteworthy that a similar microenvironment pressure has been used to explain the poor survival outcomes of patients with brain metastasis, regardless of the nature of the primary tumors [92]. Thus, the brain tumor niche is emerging as a critical regulator of cancer progression. Here we will summarize some of its main components (Figure 4).

### 4.1. Extracellular Matrix in the Brain

The brain has a distinct and very abundant extracellular matrix content, enriched in proteoglycans, glycoproteins and glycosaminoglycans. Likewise, proteoglycans such as heparan sulfate (HS) and hyaluronic acid (HA) are accumulated in certain areas, favoring the formation of vascular and stem cell niches [93,94]. The different components of this matrix favor a coordinated action between the tumor cells and their surround stroma. CD44, which is expressed mostly in glioma cells of the MES subtype, can bind HA and osteopontin (OPN) allowing tumor cells to be retained in specific niches [95,96]. Moreover, OPN is an important chemokine for the recruitment and the M2 differentiation of macrophages [97]. Heparanase (HSPE), the main endo-glucuronidase that releases the HS side-chains of heparan sulfate proteoglycans, modifies the matrix and stimulates the proliferative and invasive capacities of gliomas [98,99]. In addition, HSPE can modify the microenvironment by inducting the production of angiogenic signals [100].

Genes specifically expressed by tumor cells, like *tenascin C (TNC)*, *periostin* and *TIMP1* (*Tissue inhibitor of metalloproteinases 1*) can shape the brain parenchyma. *TNC* increases the stiffness of the extracellular matrix, which compromises the vascular integrity and induces hypoxia and HIF-1α signaling. In a feedback loop regulation, mutant IDH1 reduces HIF-1α-dependent TNC expression and decreases the aggressiveness of the gliomas [101] through changes in the endothelial cells and the microglia [102] and by promoting the acquisition of a mesenchymal stem-like phenotype by tumor cells [103]. In a similar way, periostin has been associated with the recruitment of M2 macrophages, which in turn stimulates the growth of cells with glioma stem characteristics [104]. Moreover, NFκB activation, a hallmark of the MES phenotype in gliomas, promotes the expression of a series of matrix regulators such as *Serpine1*, *Plau* (plasminogen activator, urokinase) and *TIMP1*, inducing tumor proliferation and growth [105]. Altogether, these results highlight the strong connection between glioma (particularly GBM) cells and the extracellular matrix, which is necessary for the development of this disease.

### 4.2. The Brain Vasculature

The brain has a special vasculature, characterized by an increased vessel density, a very high pericyte/endothelial ratio, and the presence of the blood brain barrier (BBB) (Figure 4). This is relevant both for the beginning and the progression of gliomas, differentiating them from the rest of the cancers. It also adds complexity to the search for therapies against these tumors. The process of vascularization of gliomas is complex and involves the participation of several factors, like Vascular Endothelial Growth Factor A (VEGFA), Angiopoietin 1/2 (ANG-1/2), Interleukin 8 (IL-8), PDGFA, Fibroblast Growth Factors (FGFs), Transforming Growth Factor Beta 1 (TGF-β1) and Hepatocyte Growth Factor (HGF) [7,106]. It has been postulated that during the first steps of gliomagenesis the tumor cells group around the pre-existing vessels (vascular co-option) to satisfy their high demand of nutrients and oxygen. This mechanism is used as well by glioma cells to invade the brain tissue and to escape from antiangiogenic treatment [107]. Later on, new vessels begin to generate from pre-existing vessels (neoangiogenesis). Hypoxia is the main inducer of this process through the activation of HIF-1α, which generates a robust angiogenic response mediated in part by the increased expression and secretion of VEGFA and IL-8 [108]. Generation of blood vessels in gliomas can be increased by various processes such as vasculogenesis, which consists in the recruitment and differentiation of endothelial progenitors from the bone marrow [109]. Hypoxia can induce this mechanism through the induction of Stromal-cell Derived Factor-1 (SDF-1) and the recruitment of CXCR4 (C-X-C Motif Chemokine Receptor 4)-expressing bone marrow-derived cells [110,111]. Another relevant process in the formation of the vasculature (especially in higher grade gliomas) is the so-called vascular mimicry, which is the ability of tumor cells to form functional vascular networks similar to vessels. Indeed, tumor cells have been shown to differentiate into endothelial cells [112,113,114] and pericytes [78], which are necessary for the sustenance and protection of the endothelium. The result of this entire angiogenic process is the formation of an aberrant vasculature, characterized by dilated and tortuous vessels with a permeable endothelium and a reduced pericyte coverage [7]. The appearance of a leaky BBB is a hallmark of the most aggressive gliomas and allows their diagnosis through the visualization of the contrast agent gadolinium extravasation by magnetic resonance imaging [115].

The antibody against VEGFA (bevacizumab) has been extensively tested in GBM patients. Although it has failed to prolong survival in newly diagnosed and recurrent tumors, it improves the progression-free survival and maintains the baseline quality of life and the performance status of treated patients so it is routinely used in the clinic [116]. Furthermore, it has been postulated that therapies inducing a vascular normalization of the tumors, rather than reducing the blood vessels, are sufficient to make them sensitive to traditional therapies and even immunotherapies [117,118]. Treatment with cediranib, a pan-VEGF receptor inhibitor showed some good initial results normalizing the vasculature and reducing the glioma-associated edema [119], although later on it has not proved to prolong significantly the survival of GBM patients [8].

More recently, results with bi-specific antibodies against ANG2 (angiopoietin 2) and VEGF [120,121], or ANG2 and TIE2 (tunica interna endothelial cell kinase 2) [115] have shown a strong therapeutic effect in animal models. These dual approaches enhance the morphological normalization of vessels and produced a change in the phenotype of the tumor´s macrophages [115,120,121]. However, they have not been tested in patients yet.

### 4.3. The Immune Cell Component of Gliomas

The brain has long been recognized as an immune privileged tissue because of the restrictions imposed by the BBB. Moreover, this tissue was supposed to lack a peripheral lymphatic system [122]. This viewpoint has been revised recently and now the brain is proposed to be an immunologically distinct rather than a privileged organ [123]. Nevertheless, lower and higher-grade gliomas are considered as “cold” tumors, due to the very limited infiltration of lymphocytes [9] and their low responses to different immunotherapy strategies [124]. Besides, brain tumors have been recently associated with peripheral immunosuppression [125]. It has been proposed that this lymphopenia is caused by the downregulation of Sphingosine-1-phosphate receptor 1 (S1PR1) expression in naïve lymphocytes, which are retained in the bone marrow [126]. However, the brain tumor signals that mediate these changes have not been uncovered so far.

Lower grade gliomas are tumors with a reduced infiltrate and very few proinflammatory signals [9,127]. The methylation increase induced by *IDH1/2* mutations transcriptionally represses the essential genes for the recruitment and induction of the immune response [128]. In addition, the oncometabolite (2-HG), generated by the mutant IDH1/2 function has been associated with a suppression of the T lymphocyte activity and the inhibition of the complement activation [129,130] (Figure 2).

It is widely accepted that the percentage of immune cells (especially of the myeloid lineage) in IDHwt GBMs is higher than in their mutant counterparts. Moreover, these tumors have a strong capacity to generate immunosuppression at various molecular and cellular levels [123,131]. GBMs contain large numbers of TAMs (tumor associated macrophages/microglia), which are considered as a hallmark of this pathology. Microglia are the brain resident monocytes. They function as macrophages and drive some immunological responses in this tissue in normal and pathological conditions [123,131]. Microglia (CD45low) differ from the peripheral monocytes derived from the bone marrow (BMDMs, CD45high), which enter the brain after disruption of the BBB in aggressive gliomas. However, these two populations are difficult to distinguish under pathological conditions [132]. TAMs secrete molecules that support the growth and the migratory properties of tumor cells. Moreover, they contribute to generate the immune and vascular microenvironment typical of aggressive gliomas [133]. They are commonly, but not solely found in the perivascular tumor niche in the vicinity of GSCs [71]. Gliomas have been shown to actively recruit TAMs by releasing several factors, including SDF-1 (CXCL12, C-X-C Motif Chemokine Ligand 12), and M-CSF (CSF-1; Colony Stimulating Factor 1). Furthermore, TAMs are a source of cytokines such as TGF-β1, STI1 (stress inducible protein 1), EGF (epidermal growth factor), IL-6 (Interleukin 6) and IL-1β (Interleukin 1 Beta) that promote growth, migration and mesenchymal transformation of the tumor cells (Figure 4) [131,133]. Another type of myeloid cells that infiltrate the glioma and usually cooperate with the growth of the tumor are the neutrophils that favor vascular recruitment and secrete molecules such as proteins from the S100 family, particularly S100A4 (S100 Calcium Binding Protein A4), to generate vascular alterations [134].

### 4.4. Glioma Immunosuppression

Gliomas are tumors with a paucity of lymphocytes and little pro-inflammatory signals. However, what really distinguishes them from other cancers is their myeloid composition. They are highly enriched in macrophages and MDSCs (myeloid derived suppressor cells), with a strong pro-tumoral capacity [131,133]. Glioma TAMs do not express T cells co-stimulatory molecules, such as CD40, CD80 or CD86 [135]. Moreover, it has been suggested that in response to molecules secreted by glioma cells (like M-CSF) there is a macrophage polarization towards the M2 phenotype, characterized by the elevated expression of CD163 and CD204. M2 macrophages express arginase-1, Interleukin 10 (IL10) and TGF-β1, which reduce the anti-tumor activity of T cells [136]. By contrast, M1 macrophage markers, such as *CD74* and *F11R*, correlate positively with the survival of glioma patients [137,138]. Alternatively, other authors have proposed that there is an accumulation of M0 (non-polarized) macrophages in these tumors [139], or that there is a ratio of M1/M2 expression markers that correlates with a good prognosis in gliomas [140].

MDSCs are a heterogeneous group of myeloid cells that expand in pathological situations such as chronic infections and cancer. The salient features of these cells are their ability to inhibit T cell function and induce the generation of regulatory T (Treg) cells, being an important therapeutic target for modulating the immune response of the tumor microenvironment [141]. In a GBM study, it was shown that MSDCs tend to accumulate as the histological grade increases. Moreover, these cells exhibit an increase activation state, with higher expression of molecules like arginase [142]. A more recent analysis using multiparametric flow cytometry and the mass cytometric TOF (time-of-flight) (CyTOF) analysis of the immune component of GBMs has confirmed that the amount of MDSCs (in the blood and in the tumor tissue) predicts poor prognosis [143]. In addition to their immunosuppressive capacity, glioma MDSCs have been associated with tumor angiogenesis because they express KDR (kinase insert domain receptor), also known as VEGFR2, which coordinates immune alterations with vascular defects, favoring the progression of the tumors [144]. Notably, some sexual divergences have been recently shown in the amount and the subtype of MDSCs present in the tumors and in the blood of GBM patients, which could account for the worse prognosis of male glioma patients. Moreover, these results suggest a sex-dependent selection of the different immunotherapies [145].

GBMs have a strong capacity to elicit qualitative and quantitative severe T-cell dysfunction. Although scarce, there is a lymphocytic infiltrate in high-grade gliomas, which include CD4+ T helper (Th), CD8+ and CD4 Tregs (CD4+CD25+Foxp3+) cells [146]. These T cells are re-educated in the immunosuppressive microenvironment of gliomas from early stages of tumor development [147]. As a result, the small population of Treg cells show a strong immune suppressing effect on the GBM microenvironment [148] and activation of CD8 cells is impaired through the expression of immune checkpoint molecules, like PDL1 (programed death-ligand 1), by tumor and myeloid cells [149,150]. This molecule binds to PD-1 (programmed cell death protein 1), expressed on T cells, inducing exhaustion and apoptosis. It is a prominent negative regulator of T cell function and a mediator of immune evasion by tumor cells. Immune checkpoint inhibitors (ICIs) targeting PD-1/PDL1 have been approved for several cancers. Despite promising preclinical results, all the recent clinical trials in recurrent and primary GBMs found no overall survival benefit in patients receiving ICIs [151]. One possible explanation for the failures of these treatments is the prominent immunosuppressive role of the myeloid compartment of gliomas. Many strategies have been explored in order to reduce the number of these cells or to reprogram them to be more inflammatory and immunogenic. CCR2 (C-C Motif Chemokine Receptor 2) is one of the main chemoattractant receptors for macrophages and MDSCs and it has a special relevance in the development of the glioma immune microenvironment. A CCR2 antagonist has shown a synergistic effect with an antibody against PD-1, generating a very long-lasting therapeutic effect in preclinical models of murine gliomas [152]. An inhibitor of CSF1R (colony stimulating factor 1 receptor), which controls the production and differentiation of TAMs, did not deplete these cells but favored their polarization towards a M1 phenotype, inducing a strong halt in the progression of gliomas [153]. This molecule is being tested as a single agent in patients with advanced solid tumors, including GBMs (ClinicalTrials.gov Identifier: NCT02829723). A very recent study has discovered that niacin (vitamin B3) treatment increased the number of TAMs in the brain and their production of the anti-proliferative INF-α14 (interferon alpha 14). The advantage of this molecule is that it could be easily translated into clinical application in combination with TMZ or other immunotherapies [154].

## 5. Genomic Alterations that Define the Microenvironment of Gliomas

The complex brain ecosystem determines the growth of gliomas and, in a feedback interplay, the molecular determinants of the different tumors shape their growing microenvironment. However, the intricate mechanisms that govern this interaction remain to be uncovered. As we previously mentioned, the presence of mutations in *IDH1/2* defines the appearance of less aggressive tumors, largely linked to their effect on the methylation profile. Recently we and others have shown that these epigenetic changes can modify the tumor microenvironment by inhibiting angiogenesis [43] or reducing the stiffness of the extracellular matrix [101]. Moreover, IDH mutations are associated with a reduced entry of immune cells [128,155]. On the contrary, multiple modifications of the microenvironment, including vascular ones, have been linked to alterations on the *EGFR* gene [34]. A series of reports have shown that EGFR signaling induces the production of angiogenic genes such as *VEGF*, *ID3* or *IL8* [156,157,158]. In many cases, EGFR seems to control the neovascularization of glioma through the activation of NFκB or SRC (sarcoma) [43,158]. Furthermore, *EGFR* alterations have been associated with the recruitment of immune cells in gliomas [159,160].

Another glioma-related gene that has a key effect on the control of the microenvironment is *NF1*, which has been associated with the recruitment of TAMs in GBM [62] and with an enriched immune transcriptome [161]. In agreement with these in silico data, a genetic mouse model of *NF1* silencing showed an accumulation of TAMs compared to tumors driven by *PDGFA* overexpression. Moreover, these two types of genetic alterations had a different impact in the architecture of the tumor vasculature [162].

Regarding the different transcriptional subtypes, plenty of evidence supports the notion that MES gliomas are associated with vascular disorders and with the accumulation of immune cells [75,82,163,164,165]. By contrast, PN tumors show a reduced immune infiltration [62,163,164]. Moreover, OLIG2 (oligodendrocyte transcription factor 2) expression, typical of this subgroup, promotes vessel co-option and preserves the BBB [166]. With the CL gliomas the situation is not that clear: some authors have suggested a reduced myeloid content in these tumors [164,165], whereas others have observed a higher recruitment of immune cells in CL gliomas [167]. However, these discrepancies could be due to difficulties in the stratification of the tumors (Figure 3).

## 6. Conclusions

Cancer can no longer be considered as a disease of hyper-proliferative cells only. The discovery of the existence of tumor stem cells and their relation with their surrounding stroma has profoundly changed this simplistic view. This is particularly true for gliomas, which grow in a very distinct microenvironment from other tissues. To add an additional level of complexity, different vascular and immune ecosystems exist in the different gliomas, but also inside any given tumor. GSCs also display inter- and intra-tumoral heterogeneities, challenging the classification of gliomas and the selection of proper treatments. The use of several –OMICS, particularly those using single-cell analysis and spatial transcriptomics, could integrate the molecular events that drive gliomagenesis with the historical evolution of the tumors, providing a better stratification of the patients. Furthermore, rational design of combinatorial treatments to target the different subclones should be mandatory if we want to achieve a clinically meaningful effect in GBMs.

Plasticity is the other main caveat when treating gliomas. The equilibrium between the different clones that exist in a given tumor is driven by their surrounding niche and can be challenged during glioma progression and in response to the different therapies. Knowing the evolutionary trajectories of glioma phenotypes and the molecular and cellular interactions responsive for these transitions would certainly help to anticipate the resistance to therapies in different patients. Besides, there is a need for more dynamic biomarkers such as novel magnetic resonance imaging (MRI), which inform on the metabolism of the tumor cells and their microenvironment, or liquid biopsies. These techniques could add a dynamic view to the current stratification of gliomas and allow us to follow their evolution.

Any amateur hunting guide would advise that hitting a moving target like gliomas cannot be achieved with a single bullet. We need to use pellet guns: combinatorial approaches targeting different subclones as well as their supportive microenvironment. Moreover, it is mandatory to know in advance the nature of the specific tumor and the trajectories that it could take in response to treatments. Only then we will get close to a precision medicine for glioma patients, which would allow us to chase this elusive cancer.

## Figures and Tables

**Figure 1 cancers-12-01622-f001:**
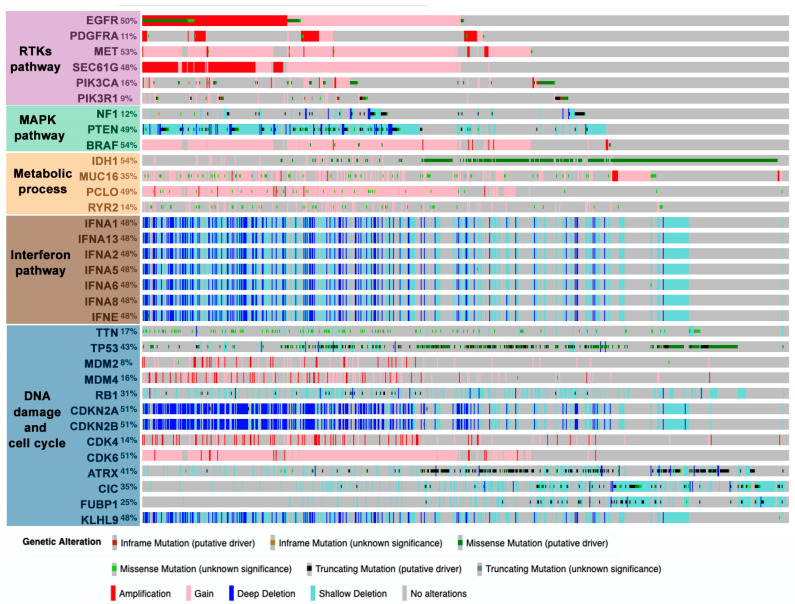
Distribution of point mutations and copy number variations in genes frequently altered in gliomas. Analysis of merged glioblastoma (GBM) and lower-grade gliomas using the TCGA dataset from cBioportal (https://www.cbioportal.org) (*n* = 794). ATRX: alpha thalassemia/mental retardation syndrome X-linked; BRAF: B-Raf proto-oncogene, serine/threonine kinase; CDK4/6: cyclin dependent kinase 4/6; CDKN2A/B: cyclin dependent kinase inhibitor 2A/B; CIC: capicua transcriptional repressor; EGFR: epidermal growth factor receptor; FUBP1: far upstream element binding protein 1; HMCN1: hemicentin 1; IDH1: isocitrate dehydrogenase 1, IFNA1/13/2/5/6/8: interferon alpha 1/13/2/5/6/8; IFNE: interferon epsilon; KLHL9: kelch like family member 9; LRP2: LDL receptor related protein 2; MDM2/4: mouse double minute 2/4; MET: proto-oncogene, receptor tyrosine kinase; MUC16: mucin 16; NF1: neurofibromin 1; PCLO: piccolo presynaptic cytomatrix protein; PDGFRA: platelet-derived growth factor receptor alpha; PIK3CA: phosphatidylinositol-4,5-bisphosphate 3-kinase catalytic subunit alpha; PIK3R1: phosphoinositide-3-kinase regulatory subunit 1; PKHD1: polycystic kidney and hepatic disease 1; PTEN: phosphatase and tensin homolog; RB1: retinoblastoma protein 1; RYR2: ryanodine receptor 2; SEC61G: Sec61 translocon gamma subunit (SEC61G); TP53: tumor protein; TTN: titin.

**Figure 2 cancers-12-01622-f002:**
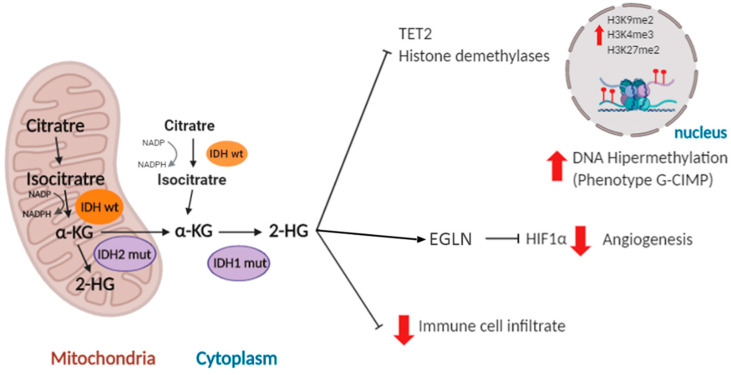
Representative image of the functions of mutant IDH1/2 in gliomas. In the presence of isocytrate dehydrogenase 1/2 (IDH1/2) mutant proteins, accumulation of 2-hydroxyglutarate (2-HG) at the expense of α-ketoglutarate (α -KG) inhibits ten-eleven translocation 2 (TET2) and histone demethylases and favors a glioma CpG island methylator (G-CIMP) phenotype. The immune and vascular microenvironment of gliomas are also affected by IDH1/2 mutations. EGLN: egl-9 family hypoxia inducible factor 1; HIF1α: hypoxia inducible factor 1, subunit alpha; NADP: nicotinamide adenine dinucleotide phosphate; NADPH: reduced nicotinamide adenine dinucleotide phosphate.

**Figure 3 cancers-12-01622-f003:**
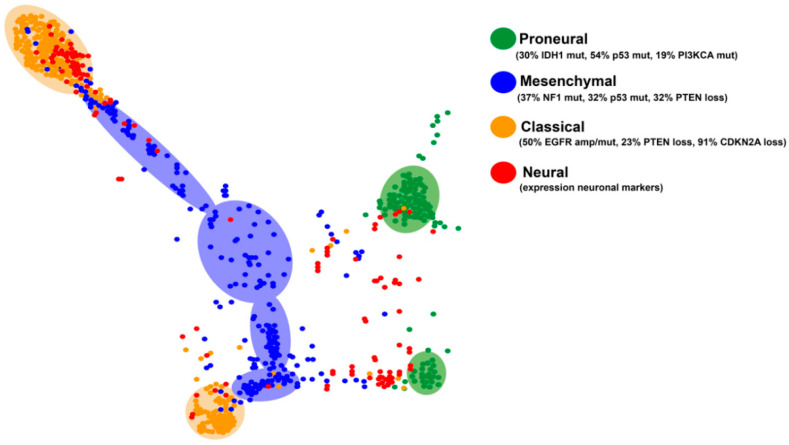
GBM clustering. Map of the different GBM subtypes and relevant genetic alterations in each one. The figure was generated by clustering tumors based on their DNA-methylation and transcriptional profiles (http://tumormap.ucsc.edu/?p=ynewton.gliomas-paper) (modified from [6,26]).

**Figure 4 cancers-12-01622-f004:**
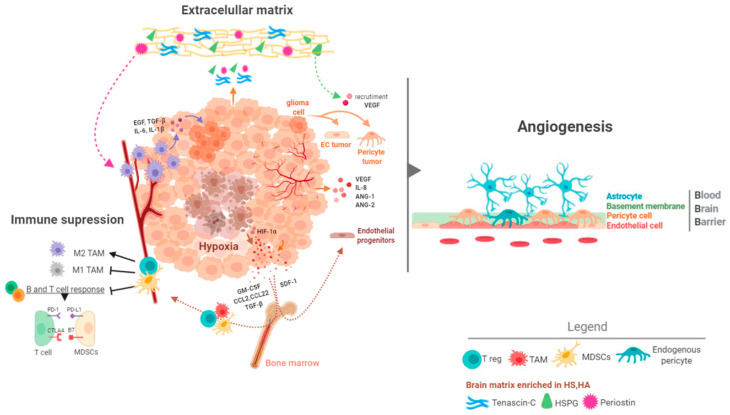
Key components of the glioma microenvironment. Glioma tissue has been described as a hypoxic and immunosuppressive microenvironment, with aberrant vasculature and impaired BBB integrity. The abundant and distinct extracellular matrix of gliomas promotes the neo-vascularization of the tumor and the recruitment of M2 macrophages. Generation of new blood vessels in gliomas is the result of endothelial proliferation, the recruitment of endothelial progenitors from bone marrow and the trans-differentiation of tumor cells into pericytes and endothelial cells. These cells form part of the BBB. Hypoxia is the main inducer of some of these processes as it favors the secretion of molecules that attract the extravasation of immune and vascular cells. Inflammatory responses are limited in the brain because the entry of peripheral immune cells is restricted by the BBB. Moreover, cytokines and chemokines secreted by glioma cells induce infiltration of immunosuppressive cells (MDSCs, Tregs and TAMs) and the acquisition of pro-tumoral phenotypes by myeloid cells: M2 phenotype differentiation and PDL-1 and B7 expression. ANG-1/2: angiopoietin 1/2; BBB: blood brain barrier; CCL2/22: C-C motif chemokine ligand 2/22; CTLA4: cytotoxic T-lymphocyte-associated protein 4; EC: endothelial cell; EGF: epidermal growth factor; HIF-1α: hypoxia inducible factor 1 alpha; HSPG: heparan sulfate proteoglycans; GM-CSF: granulocyte macrophage-colony stimulating factor; IL-6/8/1β: interleukin 6/8/1b; MDSCs: myeloid derived suppressor cells; PD1: programmed cell death protein 1; PDL-1: programmed death-ligand 1; SDF-1: stromal-cell derived factor-1; TAMs: tumor associated macrophages/microglia; TGF-β: transforming growth factor beta; Tregs: regulatory T cells; VEGF: vascular endothelial growth factor.

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
