# Peer review of "Cellular Plasticity and Tumor Microenvironment in Gliomas: The Struggle to Hit a Moving Target"

_cancers, 2020, doi:10.3390/cancers12061622_

Round 1

Reviewer 1 Report

Gargini and colleagues have reviewed for cancers the complex relationship between gliomas and their microenvironment, with emphasis on the capacity of these tumors to respond, cross-talk to the specific brain environment thanks to their plasticity. The plasticity is likely not specific of gliomas but the brain certainly has specificities, particularly in terms of immune responses.

I found this review well written, updated, and rationally designed.

Few points should be addressed.

Lines 52-3. Ref 5 shows differences in terms of treatment intensity and survival. A general statement on correlations between GBM subtypes (that are progressively shrinking from 4 to 2) and survival is not strongly supported by papers, in my opinion.

Lines 78-79. I suggest rephrasing as follows: After the 2016 revision of the WHO classification of brain tumors, the cIMPACT-NOW consortium has recently proposed a further integration of molecular data and histopathological features.

Line 170. Ref 37 demonstrates that R-2HG downregulates HIF-1 alpha trough activation of EGLn. Consequently figure 1 should show EGLn upregulation rather than. Actually previous data suggests that endostatin down-regulation increases angiogenesis (Liu et al. 2012).

Line 305. I guess the first “and” is a typo.

Line 542. “Promote” and “preserve” should be “promotes” and “preserves”.

Line 548. “Cannot longer” should be “can no longer”. “Only” could be added at the end of the sentence.

The last point is about the title. I guess the mention of the “ugly duckling of cancers” refers to glioma specificities. However, in the story, the ugly duckling becomes a beautiful swan (for a reference see  https://www.dltk-teach.com/fairy-tales/ugly-duckling/story.htm ). Thus, particularly in the case of glioblastoma, I feel the mentioning of  “ugly duckling” as considerably misleading. Even after 20 years the use of “terminator” sounds more appropriate to me (Holland 2000).

Bibliography

Holland, E.C. 2000. Glioblastoma multiforme: the terminator. Proceedings of the National Academy of Sciences of the United States of America 97(12), pp. 6242–6244.

Liu, Y., Jiang, W., Liu, J., et al. 2012. IDH1 mutations inhibit multiple α-ketoglutarate-dependent dioxygenase activities in astroglioma. Journal of Neuro-Oncology 109(2), pp. 253–260.

Author Response

Lines 52-3. Ref 5 shows differences in terms of treatment intensity and survival. A general statement on correlations between GBM subtypes (that are progressively shrinking from 4 to 2) and survival is not strongly supported by papers, in my opinion.

We agree with the referee that the initial survival benefit of Proneural tumors has been attributed to the presence of IDHmutations in this subgroup. Although there are some evidences of a predictive value of the transcriptional classification of gliomas, it is true that there are still controversies related to this issue. Thus, we have changed the sentence in the new version of the manuscript.

Lines 53-55. However, this stratification does not dictate a different therapeutic approach so it is not routinely performed in the clinic.

Lines 78-79. I suggest rephrasing as follows: After the 2016 revision of the WHO classification of brain tumors, the cIMPACT-NOW consortium has recently proposed a further integration of molecular data and histopathological features.

We thank the reviewer for this comment. Now the sentence is more precise. We have included it as suggested in lines 80-82 of the new manuscript.

Line 170. Ref 37 demonstrates that R-2HG downregulates HIF-1 alpha trough activation of EGLn. Consequently, figure 1 should show EGLn upregulation rather than. Actually previous data suggests that endostatin down-regulation increases angiogenesis (Liu et al. 2012).

We are sorry for that mistake. We have corrected the Figure 1, which is now Figure 2, and the text.

Lines 190-194. Besides, 2-HG can inhibit hypoxia inducible factor 1, subunit alpha (HIF1α) signaling through the stimulation of prolyl-hydroxylases such as EGLN [47]. In relation with this, the epigenetic changes induced by IDH mutations has been linked to changes in the microenvironment that impair the progression of gliomas. These alterations will be discussed in further detail below. 

Line 305. I guess the first “and” is a typo.

We have corrected the complete sentence.

Lines 335-337. EMT is a dynamic and reversible process in which cells lose their ability to polarize and form intercellular junctions. Instead, they acquire mesenchymal characteristics, such as elevated resistance to apoptosis and high migratory capacity.

Line 542. “Promote” and “preserve” should be “promotes” and “preserves”.

We have corrected the sentence.

Lines 575-576. Moreover, OLIG2 expression, typical of this subgroup, promotes vessel co-option and preserves the BBB [166].

Line 548. “Cannot longer” should be “can no longer”. “Only” could be added at the end of the sentence.

We have corrected the sentence.

Line 582. Cancer can no longer be considered as a disease of hyper-proliferative cells only

The last point is about the title. I guess the mention of the “ugly duckling of cancers” refers to glioma specificities. However, in the story, the ugly duckling becomes a beautiful swan (for a reference see  https://www.dltk-teach.com/fairy-tales/ugly-duckling/story.htm ). Thus, particularly in the case of glioblastoma, I feel the mentioning of  “ugly duckling” as considerably misleading. Even after 20 years the use of “terminator” sounds more appropriate to me (Holland 2000).

Bibliography

Holland, E.C. 2000. Glioblastoma multiforme: the terminator. Proceedings of the National Academy of Sciences of the United States of America 97(12), pp. 6242–6244.

Liu, Y., Jiang, W., Liu, J., et al. 2012. IDH1 mutations inhibit multiple α-ketoglutarate-dependent dioxygenase activities in astroglioma. Journal of Neuro-Oncology 109(2), pp. 253–260.

We thank the reviewer for this sharp comment. He/she is completely right. In this new version of the manuscript we have changed the title of the review: Cellular plasticity and tumor microenvironment in gliomas: the struggle to hit a moving target. We are aware that other groups have previously used the moving target expression for different cancers, including gliomas (Spencer et al., Curr Cancer Drug Targets. 2017;17(3):236-254; Nakano, J Neurosurg. 2015 Feb;122(2):324-30). However, they all referred to the elusive properties of glioma stem cells. Here we have considered the plasticity of all the components of glioma tumors.

We have also changed the Conclusions section in response to Reviewer 3. There we explain the meaning of the new title.

Line 600-605. Any amateur hunting guide would advise that hitting a moving target like gliomas cannot be achieved with a single bullet. We need to use pellet guns: combinatorial approaches targeting different subclones as well as their supportive microenvironment. Moreover, it is mandatory to know in advance the nature of the specific tumor and the trajectories that it could take in response to treatments. Only then we will get close to a precision medicine for glioma patients, which would allow us to chase this elusive cancer.

Reviewer 2 Report

This Review by Gargini and colleagues discusses recent updates to the genetic and molecular classification of gliomas, and focuses on glioma stem cells as a major contributor to therapeutic resistance and also the role of the brain tumor microenvironment, including the tumor vasculature and immune cell infiltration. The work addresses an important topic, given that glioblastoma has a poor prognosis with very few advancements in outcomes in recent decades. The paper is well written and informative, providing clinicians and researchers an accurate up-to-date overview of the state-of-play of the field and touching on future research directions. I have some comments that need to be addressed in order to improve the manuscript:

  1. In the Introduction, it is stated that low grade gliomas are hardly removable by surgery. Although the diffuse nature presents a challenge, it is worth noting that early surgical resection low grade gliomas improves outcomes by delaying transformation to GBM, citing work including: Sanai N, Berger M, 2008, Neurosurgery, Glioma extent of resection and its impact on patient outcome; Noorani I, Sanai N, Neurosurgical Clin N Am, 2017, Surgical management of incidental gliomas. 
  2. The fact that EGFR amplification portends a poorer prognosis despite histology being low grade needs citing: Stichel et al, 2018, Acta Neuropathol, Distribution of EGFR Amplification, Combined Chromosome 7 Gain and Chromosome 10 Loss, and TERT Promoter Mutation in Brain Tumors and Their Potential for the Reclassification of IDHwt Astrocytoma to Glioblastoma.
  3. The authors point out that combinatorial treatments aimed at multiple pathways in GSCs may benefit, including against EGFR. A brief mention here is needed that EGFR inhibitors have not provided clinical benefits thus far, at least partly because EGFR amplifications are found on double minute chromosomes that disappear on inhibition and reappear on removal of therapy. Moreover, multiple cooperative partners of EGFR are likely to drive gliomagenesis, only some of which are currently known eg Guo et al 2017, Nature Neurosci, A TNF–JNK–Axl–ERK signaling axis mediates primary resistance to EGFR inhibition in glioblastoma.
  4. The role of CD133 GSC targeted therapy should be described, for example recent work using CAR-T cells in animal models - Vora et al, 2020, Cell Stem Cell, The Rational Development of CD133-Targeting Immunotherapies for Glioblastoma.
  5. The authors correctly discuss the genetic evolutionary events underlying gliomagenesis, including cooperatively between EGFR and PTEN / CDKN2A for example. It is important to highlight that such knowledge has been gathered through genetically engineered mouse models of glioma, which enable cooperation between mutations to be biologically explored in the cell-of-origin. Citations of references to include here are: Noorani, 2019, Cancers, Genetically Engineered Mouse Models of Gliomas: Technological Developments for Translational Discoveries; Llaguno et al, 2019, Nature Neurosci, Cell-of-origin Susceptibility to Glioblastoma Formation Declines With Neural Lineage Restriction.

Minor comment:

- Line 548 'Cancer cannot longer' should be 'Cancer can no longer'.

Author Response

1. In the Introduction, it is stated that low grade gliomas are hardly removable by surgery. Although the diffuse nature presents a challenge, it is worth noting that early surgical resection low grade gliomas improves outcomes by delaying transformation to GBM, citing work including: Sanai N, Berger M, 2008, Neurosurgery, Glioma extent of resection and its impact on patient outcome; Noorani I, Sanai N, Neurosurgical Clin N Am, 2017, Surgical management of incidental gliomas. 

Following the reviewer suggestion, we have included a new reference and we have changed the complete paragraph in the introduction.

Lines 39-43. Graded II is assigned to slow developing tumors, whereas grade III tumors show histological symptoms of aggressiveness, such as cellular pleomorphism and mitotic activity. Both types of gliomas have an infiltrative nature and they are challenging for neurosurgeons. However, early surgical resection of these gliomas improves outcomes by delaying their transformation into higher grade tumors [3].

2. The fact that EGFR amplification portends a poorer prognosis despite histology being low grade needs citing: Stichel et al, 2018, Acta Neuropathol, Distribution of EGFR Amplification, Combined Chromosome 7 Gain and Chromosome 10 Loss, and TERT Promoter Mutation in Brain Tumors and Their Potential for the Reclassification of IDHwt Astrocytoma to Glioblastoma.

We thank the reviewer for this comment. We have cited the suggested reference in the new version of the manuscript.

Lines 104-106. In fact, detection of EGFR amplification or mutations (such as the common vIII deletion or point mutations) is associated with an increase in the aggressiveness of these gliomas, even if the histology corresponds to a grade III tumor [20].

3- The authors point out that combinatorial treatments aimed at multiple pathways in GSCs may benefit, including against EGFR. A brief mention here is needed that EGFR inhibitors have not provided clinical benefits thus far, at least partly because EGFR amplifications are found on double minute chromosomes that disappear on inhibition and reappear on removal of therapy. Moreover, multiple cooperative partners of EGFR are likely to drive gliomagenesis, only some of which are currently known eg Guo et al 2017, Nature Neurosci, A TNF–JNK–Axl–ERK signaling axis mediates primary resistance to EGFR inhibition in glioblastoma.

We agree with the reviewer that EGFR therapies have been disappointing so far in glioma patients. We have actually participated in some of the clinical attempts (new reference 31). We have included the suggested comments and references in the new version of the manuscript.

Lines 133-141. A paradigmatic example is the failure of therapies aimed at inhibiting EGFR signaling [31]. The loss of double minute chromosomes carrying EGFR amplifications during treatment and their reappearance after therapy removal could partially explain the poor therapeutic potential of the kinase inhibitors [32]. Moreover, there is an evolutionary tendency to transform linear signaling pathways into networks with multiple regulatory options and redundant pathways [1,2,33]. In the particular case of EGFR, cooperation with inhibitors of downstream molecules or other RTKs has been proposed [34]. However, only some of the EGFR partners are known in advanced and resistance often comes from the activation of unexpected pathways [35]. Novel combinatorial…

4- The role of CD133 GSC targeted therapy should be described, for example recent work using CAR-T cells in animal models - Vora et al, 2020, Cell Stem Cell, The Rational Development of CD133-Targeting Immunotherapies for Glioblastoma.

We thank the reviewer for reminding us of this seminal work. We have included a new paragraph with the suggested reference in the new version of the manuscript.

Lines 304-307. As a proof-of-concept of the relevance of stem cells in gliomas, chimeric antigen receptor T cells (CAR-T) recognizing CD133, a protein coded by the prominin I gene that is highly expressed in GSCs [63], has demonstrated superior efficacy in patient-derived GBM xenograft models [67].

5- The authors correctly discuss the genetic evolutionary events underlying gliomagenesis, including cooperatively between EGFR and PTEN / CDKN2A for example. It is important to highlight that such knowledge has been gathered through genetically engineered mouse models of glioma, which enable cooperation between mutations to be biologically explored in the cell-of-origin. Citations of references to include here are: Noorani, 2019, Cancers, Genetically Engineered Mouse Models of Gliomas: Technological Developments for Translational Discoveries; Llaguno et al, 2019, Nature Neurosci, Cell-of-origin Susceptibility to Glioblastoma Formation Declines With Neural Lineage Restriction.

Following the reviewer´s suggestion we have included in the new version of the manuscript a paragraph that mentions the evolutionary and cooperative nature of the genetic alterations commonly found in gliomas, and how we have acquired that knowledge through the study of GEMMs.

Lines 125-131. Alterations in these pathways tend to occur in an evolutionary way, with gain of chromosome 7 (EGFR, PDGFA), loss of chromosome 9/9p (CDKN2A/B), and loss of chromosome 10/10q (PTEN) occurring at early stages in gliomagenesis [27,28]. By contrast, TERT promoter mutations occur later and confer a rapid growth advantage [28]. These main drivers cooperate with each other during glioma progression. Much of the information about these interactions has been gathered through the study of genetically engineered mouse models of gliomas, which enable us to explore them in the cell-of-origin [29,30].

Reviewer 3 Report

The review was well written and focused on the important area that needs attention in the Gliomas research. The author's detailed new epigenetic insight into the Glioma from the literature. It's covered a lot about IDH1/2 mutations role in the pathology of Diffuse Glioma's. It's provided collective knowledge about the difference in the microenvironment of  Glioma compared to other tumors. However, I have some concerns which need to be addressed before considering the manuscript.

1) Authors should include a table that details i) known genetic alterations ( amplification, mutation...) and ii) predominant pathways altered in Glioma from the recent literature. Though they touched upon line 190-194; 216-218 & 119-124.

2) As pointed in lines 262-263, Glioma expresses diverse transcript programs how this going to challenge the application of the molecular classification of glioma in the treatment strategy? 

3) The conclusion part should include more information, it's very short in the review. The authors mentioned new OMICS help related to understanding tumor heterogeneity. I would suggest them to include the prospect of Single-cell Genomics and Spatial transcriptomics in understanding the Glioma tumor microenvironment and heterogeneity. 

4) Minor correction: The special characters "alpha, beta " displays the wrong symbol in the sections. Example: line 364, 368 many more.

Author Response

  • Authors should include a table that details i) known genetic alterations (amplification, mutation...) and ii) predominant pathways altered in Glioma from the recent literature. Though they touched upon line 190-194; 216-218 & 119-124.

Following the reviewer´s suggestion, we have included a new Figure 1 (lines 144-148) that depicts the distribution of the main genomic alterations found in gliomas. We have also included in the text a new sentence to mention that figure.

Lines 122-124. In general, the combination of different –OMICs has established some predominant pathways that participate in the development of gliomas: RTKs, mitogen-activated protein kinase (MAPK), metabolic processes, interferon, and DNA-damage and cell cycle regulation (Figure 1) [19,26].

  • As pointed in lines 262-263, Glioma expresses diverse transcript programs how this going to challenge the application of the molecular classification of glioma in the treatment strategy? 

We agree with the reviewer, intratumor heterogeneity challenges the stratification of gliomas. We have added a new paragraph to mention that it has been observed also at the DNA methylation level. Moreover, we now propose that a combination of gene expression, DNA methylation and genomic profiling, especially at a single-cell level, could be more robust and informative.

Lines 286-292. …., making exceedingly difficult to predict the response of a particular GBM based on its transcriptional profile. Intratumor DNA methylation heterogeneity has also been described [61], although the methylation subtypes seems to be more consistent during tumor progression [62]. Future studies with more refined and robust profiling are needed. We propose that a combination of transcriptional, epigenetic and genomic data, especially at the single-cell level, will offer a better classification of gliomas that could have a predictive value.

We have also added a new sentence to reinforce the idea of the plasticity of the transcriptional phenotypes, which adds complexity to the stratification of gliomas.

Lines 358-360. There are evidences showing that phenotypic plasticity is promoted during tumor progression and in response to therapy, with a tendency to change from PN to MES subtypes [55,86] and a frequent loss of the CL signature [62].

  • The conclusion part should include more information, it's very short in the review. The authors mentioned new OMICS help related to understanding tumor heterogeneity. I would suggest them to include the prospect of Single-cell Genomics and Spatial transcriptomics in understanding the Glioma tumor microenvironment and heterogeneity. 

We thank the reviewer for this comment. We have extended our conclusions, trying to harmonize this part of the text with the new title of the manuscript, changed following the suggestion of Reviewer 1.

Lines 582-606. Cancer can no longer be considered as a disease of hyper-proliferative cells only. The discovery of the existence of tumor stem cells and their relation with their surrounding stroma has profoundly changed this simplistic view. This is particularly true for gliomas, which grow in a very distinct microenvironment from other tissues. To add an additional level of complexity, different vascular and immune ecosystems exist in the different gliomas, but also inside any given tumor. GSCs also display inter- and intra-tumoral heterogeneities, challenging the classification of gliomas and the selection of proper treatments. The use of several –OMICS, particularly those using single-cell analysis and spatial transcriptomics, could integrate the molecular events that drive gliomagenesis with the historical evolution of the tumors, providing a better stratification of the patients. Furthermore, rational design of combinatorial treatments to target the different subclones should be mandatory if we want to achieve a clinically meaningful effect in GBMs.

Plasticity is the other main caveat when treating gliomas. The equilibrium between the different clones that exist in a given tumor can be challenged during glioma progression and in response to the different therapies. Knowing the evolutionary trajectories of glioma phenotypes and the molecular and cellular interactions responsive for these transitions would certainly help to anticipate the resistance to therapies in different patients. Besides, there is a need for more dynamic biomarkers such as novel magnetic resonance imaging (MRI), which inform on the metabolism of the tumor cells and their microenvironment, or liquid biopsies. These techniques could add a dynamic view to the current stratification of gliomas and allow us to follow their evolution.

Any amateur hunting guide would advise that hitting a moving target like gliomas cannot be achieved with a single bullet. We need to use pellet guns: combinatorial approaches targeting different subclones as well as their supportive microenvironment. Moreover, it is mandatory to know in advance the nature of the specific tumor and the trajectories that it could take in response to treatments. Only then we will get close to a precision medicine for glioma patients, which would allow us to chase this elusive cancer.

  • Minor correction: The special characters "alpha, beta " displays the wrong symbol in the sections. Example: line 364, 368 many more.

We are sorry for these typographical errors. They have been corrected throughout the text.

Round 2

Reviewer 2 Report

My comments have been addressed.

Reviewer 3 Report

Authors addressed all my concern and included suggestions in the revised version of the manuscript. I recommend the current version for the Cancers.